# Blood Adipokines/Cytokines in Young People with Chronic Bronchitis and Abdominal Obesity

**DOI:** 10.3390/biom12101502

**Published:** 2022-10-17

**Authors:** Alena Dmitrievna Khudiakova, Yana Vladimirovna Polonskaya, Victoria Sergeevna Shramko, Lilia Valeryevna Shcherbakova, Evgeniia Vitalievna Striukova, Elena Vladimirovna Kashtanova, Yulia Igorevna Ragino

**Affiliations:** Research Institute of Internal and Preventive Medicine—Branch of the Institute of Cytology and Genetics, Siberian Branch of Russian Academy of Sciences, B. Bogatkova str., 175/1, Novosibirsk 630089, Russia

**Keywords:** chronic bronchitis, abdominal obesity, population, adipokines, cytokines

## Abstract

The pathogenesis of the development of chronic lung diseases assumes the participation of systemic inflammation factors, as well as hormone-like substances produced by adipose tissue. The aim of this study was to evaluate the associations of certain adipokines/cytokines and chronic bronchitis against the background of abdominal obesity in young people. The study included 1415 people aged 25−44. In total, 115 people were selected by the random numbers method, who were divided into two subgroups: those with chronic bronchitis and abdominal obesity and those with chronic bronchitis without abdominal obesity. A control group of patients with comparable gender and age was also selected. In the group of patients with chronic bronchitis, adiponectin, TNFa and GIP levels were 1.4 times higher. The levels of C-peptide, MCP-1 and PP in the group of chronic bronchitis were 1.3 times higher compared to the control. Adipsin, lipocalin-2, IL-6 and resistin were significantly higher in the group with chronic bronchitis. Glucagon, amylin and ghrelin were 2.2, 2.3 and 3.2 times lower, respectively, in the group of patients with chronic bronchitis. Against the background of abdominal obesity, the probability of having chronic bronchitis increased with an increase in the level of lipocalin-2 and GIP and TNFa.

## 1. Introduction

Chronic bronchitis (CB) is a disease determined as the presence of cough and sputum production for at least 3 months in each of two consecutive years when other respiratory or cardiac causes of chronic productive cough are excluded. The pathomorphological basis of CB is epithelial metaplasia, accompanied by excessive mucus secretion in response to chronic inflammation of the respiratory tract [1]. The relationship between obesity and lung function is unclear. There is evidence that obesity is associated with the aggravation of the bronchial asthma course through mechanical changes in the respiratory system. Such changes include limiting the displacement of the diaphragm between the inhalation and exhalation phases. This leads to insufficient expansion of the lungs and impaired dilation of the airways. The weakening of dilation can lead to a greater contraction of the smooth muscles of the airways, which can increase the reactivity of the airways [2].

In addition to the mechanical effect of abdominal obesity on the bronchopulmonary system, the pathogenesis of chronic lung disease development assumes the participation of systemic inflammation factors, as well as hormone-like substances produced by adipose tissue [3]. Adipose tissue is an independent endocrine organ that produces adipokines that cause systemic inflammation under the influence of hypoxemia due to obesity and concomitant respiratory disorders, such as obstructive sleep apnea syndrome, chronic obstructive pulmonary disease (COPD) and hypoventilation syndrome [4].

One study reported that patients with a high body mass index (BMI) suffer from chronic cough more often than other patients [5]. A cohort study conducted in Taiwan showed that high values of obesity-related indices, such as BMI, waist-to-hip ratio, waist-to-height ratio, etc., were associated with a rapid decline in lung function over a 4-year follow-up period [6]. It is worth mentioning that all the above studies included older patients (over 54 years old). Considering that the first changes in the lungs associated with abdominal obesity are formed in early childhood, persons of working age (up to 45 years) are of particular interest for therapeutic effects. However, there are very few such studies devoted to the peculiarities of the course of CB against the background of abdominal obesity (AO). Earlier, we reported [7] that the prevalence of CB in the young population of 25–44 years is 44.2%, and CB was 1.5 times more common in AO, which indicates the extreme urgency of this problem.

The aim of this study was to evaluate the associations of some adipokines and CB against the background of AO in young people.

## 2. Materials and Methods

A population survey of the Novosibirsk population aged 25–44 years was conducted on the basis of NIITPM—a branch of ICIG SB RAS—in 2013–2016. The study was supported by the grant of the Russian Science Foundation No. 21-15-00022.

To build a population sample, the base of the Territorial Compulsory Health Insurance Fund for Persons aged 25−44 was used for one of the districts of Novosibirsk, typical in industrial, social, population-demographic, transport structures and the level of migration of the population. A random representative sample of 2500 people was formed using a random number generator. Young age groups are known to be among the most rigid in terms of response; therefore, methods of phased epidemiological stimulation were applied: mail invitations, phone calls and information messages in the media. In total, 1512 people were examined at the screening; the response was 60.5%. After excluding pregnant women and women on maternity leave from the study, 1415 people were included in the study, including 670 men (47.3%) and 745 women (52.7%). The average age of the surveyed was 37.3 [31.8; 42.0] years. Informed consent was obtained from all persons for the examination and processing of personal data.

Chronic bronchitis (CB), according to the clinical recommendations of the Russian Respiratory Society of 2021, was established on the basis of anamnesis: the presence of cough and sputum production for at least 3 months in each of two consecutive years in the absence of other pathology of the bronchopulmonary system (tuberculosis, bronchiectasis, pneumonia, asthma, lung cancer, etc.), causing “cough history”. Abdominal obesity (AO) was determined based on waist circumference > 80 cm in women and > 94 cm in men [8].

From a population sample of patients with CB (303 people), 115 people were selected by random numbers, which were divided into two subgroups: with CB and AO and with CB without AO. A control group of patients without a history of CB was also selected, comparable in gender and age, which was also divided into subgroups: persons with AO and conditionally healthy persons (Figure 1).

A team of doctors trained in standardized epidemiological methods of screening examinations conducted the screening. The survey program included: demographic and social data, a survey on smoking habits and alcohol consumption, a socio-economic survey, a dietary survey, a history of chronic diseases and medication use, a Rose Angina Questionnaire, anthropometry, the threefold measurement of blood pressure (BP), spirometry, ECG recording with transcription according to the Minnesota code and others.

Blood pressure was measured three times with an interval of two minutes on the right hand in a sitting position after a 5 min rest using an Omron M5-I automatic tonometer with the registration of the average value of three measurements. Arterial hypertension (AH) was recorded at systolic blood pressure (SAD) ≥140 mmHg and/or diastolic blood pressure (DAD) ≥90 mmHg.

The calculation of the body mass index (BMI) was carried out according to the formula: body weight (kg) divided by the square of height (m^2^). Elevated BMI was considered >25 kg/m^2^. According to the 2019 ESC/EAS guidelines, physical activity was considered sufficient if more than 3.5 h of physical activity per week.

Those who smoked at least one cigarette a day were considered smokers.

A single blood sampling from the ulnar vein was performed after 12 hours’ night fasting. Blood parameters of lipid profile, glucose and creatinine were measured by an enzymatic method using standard ThermoFisher reagents on an automatic biochemical analyzer Konelab 30i (Thermo Fisher Scientific, Joensuu, Finland). The conversion of serum glucose into plasma glucose was carried out according to the formula: plasma glucose (mmol/L) = −0.137 + 1.047 × serum glucose (mmol/L). Elevated blood levels of LDL-C were considered ≥116 mg/dL, elevated blood levels of Non-HDL-C were considered ≥130 mg/dL and elevated blood levels of TG were considered ≥150 mg/dL.

The levels of amylin, C-peptide, ghrelin, glucose-dependent insulinotropic polypeptide (GIP), glucagon, interleukin 6, insulin, leptin, monocytic chemotactic factor 1 (MCP-1), pancreatic polypeptide (PP) and tumor necrosis factor-alpha (TNF-α) were determined by multiplex analysis using the Human Metabolic Hormone V3 (MILLIPLEX) panel. The Human Adipokine Magnetic Bead Panel 1 was used to determine the levels of adiponectin, adipsin, lipocalin-2, plasminogen activator inhibitor type 1 (PAI-1) and resistin.

Statistical processing of the obtained results was carried out using the SPSS software package (version 13.0, IBM, New York, NY, USA). The normal distribution was checked using the Kolmogorov–Smirnov criterion. Due to the nonparametric distribution of the studied indicators, the data are presented for continuous variables in the form of Me [25, 75], where Me is the median and 25 and 75 are the first and third quartiles in the case of categorical variables in the form of absolute and relative values: n (%). The nonparametric Mann–Whitney U-test was used to compare two independent samples. Pearson’s chi-squared criterion was used to compare the fractions. Associations were evaluated using multiple logistic regression analysis performed under the following conditions: the dependent variable is dichotomous: the presence/absence of CB; the independence of observations; the absence of multicollinearity, i.e., situations where independent variables strongly correlate with each other (r > 0.7); linear dependence between each independent variable and the logarithm of the odds ratio (logarithmic coefficients); the independence of residuals. The results of multiple logistic regression analysis were presented as OR and 95% CI for OR. The critical significance level of the null hypothesis (*p*) was assumed to be 0.05.

## 3. Results

At the first stage of our study, the clinical and anamnestic data of patients with CB and the control group were analyzed. Patients with CB had a higher heart rate (HR) and BMI and higher values of waist circumference (WC) and hip circumference (HC) than patients without CB. In addition, patients with CB spent less time per week on physical activity (PA) (Table 1). Additionally, in the group of patients with CB, smoking was registered two times more often than in the group of patients without CB (57.0% vs. 27.8%, *p* = 0.0001).

In the group of patients with CB, SAD was 1.1 times higher in the subgroup with AO (123.8 [116.3;135.9] vs. 116.5 [106.3;127.0] mmHg, *p* = 0.003), DAD was 1.3 times higher (81.00 [74.5;91.1] vs. 77.5 [70.0;84.3] mmHg, *p* = 0.047) and the triglyceride level was 1.4 times higher (98.5 [78.5;165.8] vs. 68.0 [47.0;106.5] mg/dL, *p* = 0.001) compared with patients without AO.

The control group also had higher rates of SAD and DAD (128.0 [116.4;139.0] vs. 118.5 [105.0;126.5] mmHg, *p* = 0.0001 and 87.8 [78.3;91.9] vs. 76.0 [70.0;84.3] mmHg, *p* = 0.0001, respectively) in the group with AO. Higher lipid values were also registered (cholesterol 6.1 [4.3;7.5] vs. 5.8 [3.6;6.5] mg/dL, *p* = 0.028; cholesterol-non-HDL 203.5 [109.0;233.0] vs. 175.0 [81.0;196.0] mg/dL, *p* = 0.001; triglycerides (TG) 141.5 [76.8;215.0] vs. 88.5 [53.1;115.1] mg/dL, *p* = 0.0001) and blood glucose (6.1 [4.3;7.5] vs. 5.8 [3.6;6.5] mmol/L, *p* = 0.001) in the subgroup with AO than that without it.

There were no statistically significant differences in the frequency of risk factors in the group of examined patients with CB with AO and without it. A statistical trend was obtained for hypertriglyceridemia. When evaluating the same indicators depending on gender, it was found in men that hypertriglyceridemia is three times more likely to be registered in the presence of AO. No similar data were obtained for women (Table 2).

The logistic regression analysis of the chance of having CB in the young4 population of Novosibirsk showed that CB was almost 3.5 times more common in smokers than in non-smokers and almost 4.5 times more common in the presence of AO than without it (Table 3).

The next stage was the study of the adipokine levels in the examined groups. In the group of patients with CB, adiponectin, TNFa and GIP levels were 1.4 times higher. The levels of C-peptide, MCP-1 and PP in the CB group were 1.3 times higher compared to the control. The indicators of adipsin (1.5 times), lipocalin-2 (1.9 times), IL-6 (2.5 times) and resistin (4.2 times) were significantly higher in the CK group. Additionally, the three studied molecules showed lower values compared to the control group. Glucagon was 2.2 times lower in the group of patients with CB, amylin 2.3 times and ghrelin 3.2 times. (Table 4).

When assessing the concentrations of the studied molecules in the subgroups with AO, we revealed that with a combination of CB and AO, the level of insulin was 1.4 times higher, the level of PAI-1 was 1.5 times higher, the level of C-peptide was 2 times higher and the level of leptin was almost 3 times higher compared to the subgroup with CB and without AO.

In turn, the control group with AO also had higher levels of insulin, PAI-1, C-peptide and leptin (1.4, 1.2, 1.3 and 2.4 times, respectively), as well as higher values of IL-6 and TNFa compared with those without AO (Table 5).

The results of subsequent logistic regression analysis (standardization by age, gender and smoking status) of the association of adipokines with the chance of having CB in all young people included in the study are presented in Table 6. An increase in the level of GIP by 1 pg/mL is associated with an increase in the chance of having CB by 6%. With an increase in TNFa by 1 pg/mL, the chance of having CB increased by 86%. An increase in the level of lipocalin -2 by 1 mcg/L was associated with an increase in the chance of having CB by 0.1%. In addition, the chance of having CB decreased with an increase of 1 pg/mL of amylin and ghrelin by 12 and 1%, respectively.

Against the background of AO, the chance of having CB increased with an increase in the level of lipocalin-2 by 1 mcg/mL and an increase in the level of GIP by 1 pg/mL by 1% and 4.6%, respectively. It was also found that the chance of having CB with an increase in the level of TNFa by 1 pg/mL increases almost two times, but only a statistical trend has been achieved. It is worth noting that smoking included in the model increased the chance of having CB by almost seven times (Table 7).

## 4. Discussion

The relationship between the presence of abdominal obesity and impaired respiratory function has been traced since early childhood and is most often associated with the development of bronchoobstructive syndrome [9]. Among children of the indigenous population of Canada, obesity turned out to be one of the main predictors of the development of chronic bronchitis in children under 17 years of age, along with risk factors such as smoking by parents in signs of mold and mildew in the place of residence [10].

The literature mainly presents data on the effect of abdominal obesity on the course of chronic obstructive pulmonary disease (COPD), when the changes become irreversible. In one study, it was shown that with the development of chronic bronchitis against the background of COPD, patients are less likely to register an increase in BMI, but at the same time, body weight deficiency is much more common (BMI 18.5 kg/ m^2^). In general, in the study, patients with low BMI and COPD had worse respiratory function and more frequent exacerbations of chronic bronchitis than patients with a BMI of ≥25 kg/m^2^ [11]. Similar data were obtained in the KOCOSS study, which included 2694 patients with COPD (mostly men (92%)) [12]. However, there are also opposite world data indicating that the presence of CB is associated with higher BMI [13,14,15]. In a large cohort study, it was shown that indicators of respiratory function were linearly and inversely related to the waist–hip ratio in both men and women. This relationship persisted after adjusting for age, BMI, smoking, physical activity index, the prevalence of bronchitis/emphysema and the prevalence of asthma [16]. In our study, it was also found that people suffering from CB also had higher BMI, WC, WH and PA, which was lower in people without CB.

There are very few studies aimed at studying adiponectin in CB.

It is known that obesity is associated with increased inflammation and hyperreactivity of the respiratory tract, oxidative stress, expression of induced nitric oxide synthase and elevated levels of nitric oxide. On the other hand, obesity is characterized by a decrease in the level of adiponectin, which acts as an anti-inflammatory and antioxidant mediator, reducing the severity of allergic asthma [17,18]. In a number of experiments on laboratory mice, it was shown that adiponectin can mitigate the course of bronchial asthma occurring against the background of obesity and that its level significantly decreases with the hyperreactivity of the bronchi associated with obesity [19,20]. In a study that included 13 infants with bronchiolitis, it was shown that the severity of viral bronchiolitis in infancy may be associated with the profile of adipokines, in particular with the leptin/adiponectin ratio, but not with obesity [21]. We found that the level of adiponectin is higher in individuals with CB, but there were no differences between groups with and without AO, as well as when adiponectin was included in regression analysis models.

There are no studies aimed at studying adipsin in CB and AO. There is evidence that adipsin can be detected in the bronchoalveolar lavage of healthy non-smokers and people with normal body weight [22], which may allow us to assume that a change in its concentration may indicate pathological changes in the bronchial tree. In a study conducted in Finland among men with moderate or severe occupational exposure to asbestos, it was shown that the level of adipsin was associated with the degree of parenchymal fibrosis, impaired lung diffusion capacity and inflammatory activity [23]. We found that the serum adipsin level in individuals with CB was 1.5 times higher than in individuals without CB; however, no differences were obtained depending on the presence of AO.

The level of lipocalin-2 in the blood serum is closely related to the total body fat content. In addition, the volume of visceral adipose tissue is an independent factor determining the levels of lipocalin-2 [24]. There have been practically no studies of this biomarker in patients with bronchitis. There is evidence of a decrease in the level of lipocalin-2 in patients with COPD compared with healthy smokers [25]. In a small study performed among people with chronic cough (more than 1 month), there were no differences in the level of lipocalin-2 compared to the control group [26]. At the same time, it was shown in experimental animal models that the level of lipocalin-2 increases with allergen sensitization and respiratory tract provocation in lung tissues and decreases in bronchoalveolar lavage fluids [27]. Based on this, we can judge the contribution of lipocalin-2 to the development of bronchial hyperreactivity. We found that with CB, the concentration of lipocalin-2 is almost two times higher than without CB. Moreover, an increase in the level of lipocalin-2 against the background of AO increases the chance of having CB by 1%.

Amylin is a peptide synthesized not only in the beta cells of pancreatic islets but also in small amounts in other organs, such as the intestinal and stomach mucosa, the lungs and the central nervous system [28]. It has been shown that an increase in BMI by 5 kg/m^2^ is associated with a twofold increase in amylin levels [29]. Additionally, a number of studies have shown that an increased level of amylin has a protective effect on lung function when exposed to the inhalation of solid particles [30,31]. We have found that the level of amylin is significantly lower in patients with CB than without it. At the same time, the chance of having CB decreases with a decrease in the level of amylin, which corresponds to the limited literature data given above. However, when evaluating this adipokine in patients with a combination of AO and CB, these patterns were not obtained.

Obesity and insulin resistance are closely intertwined [32]. An increase in insulin levels can increase bronchospasm by stimulating the Vagus nerve and blocking the inhibitor of M2-muscarinic receptors without any effect on the sensitivity to acetylcholine of smooth muscles [33]. In a study on mice, it was shown that insulin resistance caused by obesity can increase the expression of TGF-β1, pulmonary fibrosis and the hyperreactivity of the respiratory tract [34]. In turn, C-peptide is a component of the secretion of the endocrine part of the pancreas and is an indicator of insulin production. According to a small study, C-peptide may increase with a combination of obesity and atopy in women [35]. In addition, there is evidence that the concentrations of insulin, C-peptide, leptin and the glucose/insulin ratio are associated with recurrent wheezing in preschool children [36]. A number of studies indicate an increase in the level of C-peptide in patients with COPD [37,38,39]. At the same time, none of the studies evaluated the level of C-peptide and insulin in individuals with combined pathology of AO and bronchitis. We found that the level of C-peptide increases significantly in the presence of AO, especially in the presence of CB in the subject. However, the level of insulin is more likely associated with AO, since its increase was recorded regardless of the presence or absence of CB.

The level of leptin in blood plasma adjusted for visceral fat does not differ between patients with COPD and healthy people [40] and does not correlate with BMI. At the same time, in patients with CB, the concentration of leptin in plasma correlates with BMI [41]. A study conducted in China showed that a mutation in the leptin gene can contribute to the progression of CB by inhibiting its biological action [42]; hence, it can be concluded that a change in the concentration of this adipokine may reflect the presence and severity of CB. In our study, the level of leptin was significantly higher in individuals with AO, regardless of the presence or absence of CB. However, its level did not differ in the group with and without CB.

There are a huge number of studies devoted to the study of MCP-1 in obese patients, which is the subject of a number of literary reviews published over the past decade [43,44]. MCP-1 along with other inflammatory cytokines, such as TNFa, IL-6 and IL-1, can affect immune cells, leading to local and generalized inflammation [45]. A number of studies published in the last century indicate an increase in the level of MCP-1 in bronchoalveolar flushes in patients with various bronchopulmonary pathology compared with healthy people [46,47]. When assessing the level of MCP-1 in patients with chronic cough, it was determined that its concentration in sputum is many times higher than the values obtained in conditionally healthy individuals [48,49]. However, the level of MCP-1 in the blood of patients with CB has not been determined. We found that the level of MCP-1 in the blood is higher in patients with CB than without it, but there were no differences depending on the presence or absence of AO.

Similar patterns were obtained for TNFa, the level of which was higher in patients with CB but did not differ in the AO subgroups. However, an increase in TNFa increased the chance of having CB in individuals with AO. In inflammatory diseases of the bronchopulmonary system, the level of TNFa in sputum increases significantly [50]. In one of the studies conducted in Russia, it was shown that the level of TNFa in the blood was several times higher in patients suffering from COPD than in the control group and also demonstrated a reliable positive relationship between the level of TNFa, shortness of breath and cough on a visual analogue scale [51]. TNF-α is a pro-inflammatory cytokine that participates in lipid metabolism and insulin signaling in adipose tissue; therefore, its level is elevated in obese people and decreases with weight loss [52]. However, there is evidence from a recent large study conducted in Zanzibar that the level of TNF-α does not significantly differ between patients with abdominal obesity and has no associations with any of the obesity indices [53].

A number of studies have shown that the level of GIP is associated with AO regardless of the level of insulin and can be considered as a promising target for the treatment of patients with impaired carbohydrate metabolism [54,55]. At the same time, the concentration of GIP in the blood decreases with a decrease in body weight [56]. There is practically no data for changes in the level of GIP in bronchopulmonary pathology, in particular for CB. A study conducted on laboratory animals showed that dose-dependent GIP inhibited cAMP production in lung fibroblasts [57]. There is also evidence that GIP receptors are widely expressed in tumors of the pancreas, ileum and bronchi [58]. This may indirectly indicate the possibility of changing this hormone in lung diseases. According to the data we obtained, GIP was not only almost 1.5 times higher in patients with CB but also increased the chance of having CB both in the general population and in individuals with AO.

Ghrelin acts as an anti-inflammatory factor protecting against endotoxic shock by inhibiting the expression of pro-inflammatory cytokines by activated monocytes and endothelial cells and also promotes the release of growth hormone. Due to these properties, ghrelin administration suppresses the inflammation of the respiratory tract by reducing the accumulation of neutrophils in the lungs and increasing body weight [59]. According to a number of studies, the level of ghrelin in plasma is higher in patients with insufficient body weight and pathology of the bronchopulmonary system than with normal body weight [60,61]. The same trend is also determined with respect to IgE, an increased level of which may indicate the development of allergic reactions, including bronchial obstruction [62]. The data obtained by us are consistent with the data of studies that exist at the moment. Ghrelin levels were significantly lower in patients with CB, and a decrease in ghrelin levels by 1 pg/mL was associated with an increased chance of having CB in the general population but not in patients with AO. We did not receive differences in ghrelin levels in patients with and without AO. Most likely, this is due to a small number of patients with insufficient body weight, which, as mentioned above, is associated with an increase in ghrelin levels.

The main physiological function of glucagon is to maintain glucose homeostasis in cases of hypoglycemia [63]. Glucagon also has an effect on the contraction of the smooth muscles of the respiratory tract and the inflammatory response [64]. In a study on laboratory animals, it was shown that the intranasal administration of glucagon inhibits airway obstruction and reduces bronchial tree hyperreactivity and bronchoalveolar inflammation [65,66]. In turn, the level of glucagon increases with obesity, and its concentration varies in a wide range depending on the nature of the food consumed [67]. According to our data, the level of glucagon was significantly lower in patients with CB than without it, regardless of the presence or absence of AO, which corresponds to the literature data.

PP reduces food intake and also increases energy consumption, as evidenced by increased activity of sympathetic nerves innervating adipose tissue and increased oxygen consumption [68]. A number of studies indicate that blocking glucagon receptors may lead to a decrease in body weight in obese patients [69,70], so we can assume that a change in the concentration of PP in the blood may be associated with obesity. However, according to our data, the PP level did not differ in people with and without AO but was 1.3 times higher in patients with CB. We have not been able to find studies in the world literature aimed at studying PP in diseases of the respiratory system, which makes the data we have obtained especially valuable.

## 5. Conclusions

The results obtained allow us to better understand the pathogenesis of the development of inflammatory diseases of the bronchi against the background of abdominal obesity. The most promising for clinical and scientific purposes are such adipokines as GIP, lipocalin-2, ghrelin, amylin and TNFa. Taking into account the peculiarity of the sample (25–44 years old), the results obtained can be used to develop new approaches to the diagnosis and treatment of patients with CB, which can significantly reduce the complications of this disease, as well as the early disability of people of working age.

## Figures and Tables

**Figure 1 biomolecules-12-01502-f001:**
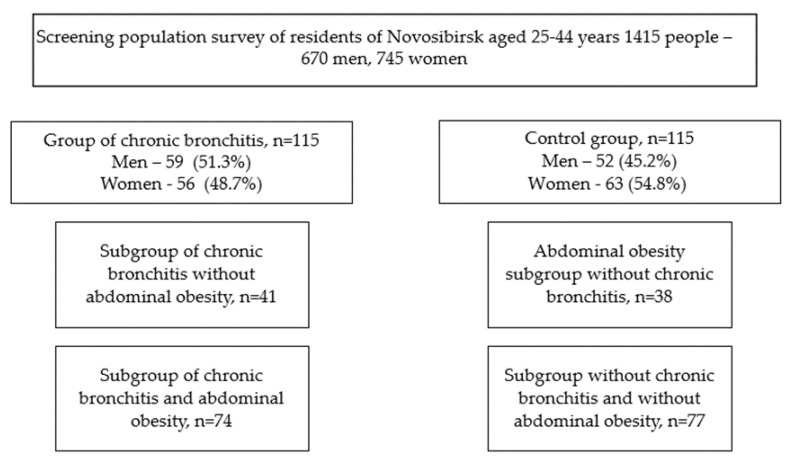
Research design.

**Table 1 biomolecules-12-01502-t001:** Clinical and anamnestic data of patients with CB and the control group.

Parameter	Group 1 CB (-), n = 115		Group 2 CB (+), n = 115		*p*
	Me	[25%;75%]	Me	[25%;75%]	
Age, years	37.0	31.8;40.8	37.9	32.3;42.2	0.135
BMI, kg/m^2^	24.8	21.8;28.6	27.1	23.5;31.4	0.004
WC, cm	84.2	74.0;93.8	90.0	81.0;101.0	0.0001
HB, cm	100.8	95.0;105.4	104.1	98.0;113.0	0.0001
SBP, mmHg.	120.0	107.0;130.0	120.0	114.0;132.5	0.277
DAD, mmHg.	79.5	70.5;88.0	79.5	73.5;88.5	0.285
HR, beats/min	72.0	64.0;78.0	74.0	67.0;82.0	0.017
PA, h/week	1.0	0.0;1.0	0.0	0.0;2.0	0.017
TCH, mg/dL	232.0	143.1;255.2	204.9	174.0;235.9	0.252
LDL-HC, mg/dL	158.5	77.3;174.0	132.2	101.6;160.0	0.366
Non-HDL-HC, mg/dL	181.0	85.0;208	161.0	124.0;188.0	0.454
TG, mg/dL	97.4	62.0;150.5	89.0	67.0;138.0	0.968
Glucose, mmol/L	5.7	5.3;6.2	5.7	5.4;6.2	0.473

Footnote: CB-chronic bronchitis, SBP-systolic blood pressure, DAD-diastolic blood pressure, AH-hypertension, HR-heart rate, PA-physical activity, BMI-body mass index, WC-waist circumference, HC-hip circumference, THC-cholesterol, LDL-HC-cholesterol low-density lipoproteins, Non-HDL-HC-cholesterol, non–high-density lipoproteins, TG-triglycerides, Me-median, 25% and 75%-first and third quartiles.

**Table 2 biomolecules-12-01502-t002:** The presence of risk factors in persons with CB, depending on the presence of AO.

Parameter		No AO (n = 41)	with AO (n = 74)	*p*
Smoking	Both sexes	22 (53.7%)	43 (58.1%)	0.561
	Men	15 (68.2%)	24 (64.9%)	0.795
	Women	7 (36.8%)	20 (54.1%)	0.222
BP ≥ 140/ ≥ 90 mmHg.	Both sexes	7 (17.1%)	20 (27.0%)	0.228
	Men	5 (22.7%)	14 (37.8%)	0.230
	Women	2 (10.5%)	6 (16.2%)	0.565
PA. ˂ 3.5 h a week	Both sexes	29 (70.7%)	58 (79.5%)	0.293
	Men	14 (63.6%)	27 (75.0%)	0.356
	Women	15 (78.9%)	31 (83.8%)	0.655
LDL-HC ≥ 116 mg/dL	Both sexes	30 (73.2%)	47 (63.5%)	0.292
	Men	15 (68.2%)	25 (67.6%)	0.961
	Women	15 (78.9%)	22 (59.5%)	0.145
Non-HDL-HC ≥ 130 mg/dL	Both sexes	30 (73.2%)	53 (71.6%)	0.859
	Men	15 (68.2%)	30 (81.1%)	0.260
	Women	15 (78.9%)	23 (67.9%)	0.203
TG ≥ 150 mg/dL	Both sexes	5 (12.2%)	20 (27.0%)	0.051
	Men	3 (13.6%)	15 (40.5%)	0.030
	Women	2 (10.5%)	5 (13.5%)	0.749

Footnote: AO-abdominal obesity, BP-blood pressure, PA-physical activity, BMI-body mass index, LDL-HC-cholesterol low-density lipoproteins, Non-HDL–HC-cholesterol non-high-density lipoproteins, TG-triglycerides.

**Table 3 biomolecules-12-01502-t003:** The relative chance of having CB associated with CB risk factors in all examined individuals. The logistic regression analysis. (Standardization by gender, age).

Parameter	OR	95% Confidence Interval (CI)		*p*
		Lower Bound	Upper Bound	
Smoking, smoking vs. not smoking	2.952	1.589	5.485	0.0001
AO, yes vs. no	3.069	1.700	5.540	0.0001
Glucose ≥ 6.1 mmol/L vs. 6.1 mmol/L	2.658	0.514	13.729	0.243
PA < 3.5 h/week vs. ≥3.5 h/week	1.466	0.769	2.794	0.245
Non-HDL-HC ≥ 130 mg/dL vs. <130 mg/dL	1.483	0.783	2.808	0.227

Footnote: AO-abdominal obesity, AH-arterial hypertension, PA-physical activity, LDL-HC-cholesterol low-density lipoproteins, Non-HDL-HC-cholesterol non–high-density lipoproteins, TG-triglycerides, OR-odds ratio.

**Table 4 biomolecules-12-01502-t004:** Levels of human adipokines studied (Me [25%;75%]).

Adipokines	Persons without CB (n = 121)	Persons with CB (n = 115)	*p*
Adiponectin, mcg/mL	32.9 [24.5;71.0]	44.7 [28.9;10.4]	0.035
Adipsin, mcg/mL	8.0 [5.5;13.9]	12.0 [8.4;17.5]	0.0001
Lipocalin-2, mcg/mL	202.8 [120.0;384.0]	392.1 [244.1;526.8]	0.0001
PAI-1, mcg/mL	18.3 [11.7;26.6]	21.2 [12.2;30.8]	0.124
Resistin, mcg/mL	36.9 [12.1;106.3]	155.1 [108.6;196.5]	0.0001
Amylin, pg/mL	13.1 [8.4;18.2]	5.7 [2.9;9.4]	0.0001
IL-6, pg/mL	0.8 [0.7;2.9]	1.2 [0.8;1.9]	0.203
C-peptide, pg/mL	815.2 [472.1;1142.1]	1011.1 [632.9;1528.4]	0.019
Insulin, pg/mL	429.8 [272.9;719.1]	390.4 [277.4;574.5]	0.132
Leptin, ng/mL	3.5 [1.5;6.5]	4.7 [1.8;9.2]	0.178
MCP-1, pg/mL	140.8 [108.3;215.3]	189.0 [128.6;276.9]	0.0001
Ghrelin, pg/mL	86.8 [29.2;154.9]	26.9 [15.8;47.6]	0.0001
TNFa, pg/mL	3.2 [2.2;4.5]	4.6 [3.2;5.8]	0.0001
GIP, pg/mL	23.0 [15.9;39.1]	32.8 [21.0;53.0]	0.002
Glucagon, pg/mL	24.2 [14.0;35.3]	10.9 [6.9;27.8]	0.0001
PP, pg/mL	34.5 [17.4;53.7]	46.5 [28.8;77.9]	0.0001

Footnote: PAI-1-type 1 plasminogen activator inhibitor, IL-6-interleukin 6, MCP-1-monocytic chemoattractant protein type 1, TNFa-tumor necrosis factor-alpha, GIP-glucose-dependent insulinotropic polypeptide, PP-pancreatic polypeptide, Me-median, 25% and 75%-first and third quartiles.

**Table 5 biomolecules-12-01502-t005:** The levels of the studied human adipokines on the background of AO and without it (Me [25%;75%]).

Adipokines	Persons without CB (n = 121)		*p*	Persons with CB (n = 115)		*p*
	without AO (n = 83)	with AO (n = 38)		without AO (n = 41)	with AO (n = 74)	
Adiponectin, mcg/mL	33.5 [24.0;6.5]	32.5 [24.5;75.5]	0.755	43.9 [33.8;12.0]	44.7 [25.1;87.2]	0.367
Adipsin, mcg/mL	7.9 [5.4;14.7]	8.0 [5.6;14.7]	0.731	12.0 [8.3;16.9]	11.9 [8.9;18.5]	0.607
Lipocalin-2, mcg/mL	187.4 [1128.0;437.3]	224.6 [130.9;383.5]	0.759	373.1 [229.5;548.7]	407.2 [259.7;514.9]	0.701
PAI-1, mcg/mL	17.3 [9.9;24.6]	20.5 [14.6;31.7]	0.034	14.9 [10.2;29.3]	23.0 [15.4;31.8]	0.017
Resistin, mgk/mL	32.5 [11.5;98.7]	52.1 [18.8;106.2]	0.445	15.9 [10.8;20.2]	15.4 [10.7;19.5]	0.881
Amylin, mcg/mL	13.4 [8,6;18,1]	12.8 [8.4;23.8]	0.897	5.8 [1.5;8.6]	5.6 [3.2;10.4]	0.232
IL-6, pg/mL	0.8 [0.6;0.9]	1.9 [0.8;3.8]	0.019	1.0 [0.7;1.6]	1.3 [0.8;2.2]	0.105
C-peptide, pg/mL	715.7 [463.4;958.2]	959.8 [570.7;1694.1]	0.006	710.8 [482.8;860.3]	1426.4 [848.8;1696.4]	0.0001
Insulin, pg/mL	406.9 [205.0;584.6]	584.6 [406.9;1059.9]	0.008	319.6 [205.7;423.6]	438.6 [277.4;672.5]	0.016
Leptin, ng/mL	2.5 [1.2;5.1]	6.0 [3.5;15.1]	0.0001	1.6 [0.7;4.3]	4.1 [2.6;5.2]	0.0001
MCP-1, pg/mL	142.1 [109.1;220.0]	139.1 [105.3;190.5]	0.661	194.5 [121.5;273.1]	186.2 [139.2;280.7]	0.369
Ghrelin, pg/mL	76.7 [28.3;155.4]	86.8 [29.1;154.6]	0.787	20.6 [13.8;55.6]	28.2 [18.8;47.6]	0.239
TNFa, pg/mL	2.9 [1.9;4.1]	4.0 [2.5;5.0]	0.046	4.1 [2.9;5.4]	4.6 [3.7;6.2]	0.070
GIP, pg/mL	21.5 [16.3;37.6]	27.3 [13.4;48.2]	0.829	34.6 [18.7;47.6]	32.8 [22.0;54.8]	0.963
Glucagon, pg/mL	20.6 [10.3;41.8]	16.8 [6.8;31.1]	0.958	10.8 [6.8;31.1]	10.9 [6.9;27.6]	0.780
PP, pg/mL	32.0 [17.3;52.6]	35.6 [17.3;56.9]	0.887	46.3 [25.6;74.9]	47.3 [30.3;79.3]	0.649

Footnote: PAI-1-type 1 plasminogen activator inhibitor. IL-6-interleukin 6. MCP-1-monocytic chemoattractant protein type 1, TNFa-tumor necrosis factor-alpha, GIP-glucose-dependent insulinotropic polypeptide, PP-pancreatic polypeptide, Me-median, 25% and 75%-first and third quartiles.

**Table 6 biomolecules-12-01502-t006:** Results of logistic regression analysis of associations of adipokines with the chance of having CB with standardization by sex, age and smoking status.

Parameter	Logistic Regression Analysis			
	OR	95% Confidence Interval (CI)		*p*
		Lower Bound	Upper Bound	
WC, cm	0.969	0.913	1.030	0.313
C-peptide, pg/mL	1.000	0.999	1.001	0.877
GIP, pg/mL	1.059	1.022	1.097	0.002
MSP-1, pg/mL	0.997	0.989	1.004	0.412
PP, pg/mL	0.997	0.980	1.013	0.693
TNFa, pg/mL	1.852	1.154	2.972	0.011
Adipsin, mcg/mL	1.000	0.924	1.081	0.994
Lipocalin-2, mcg/mL	1.004	1.001	1.008	0.016
Amylin, pg/mL	0.882	0.807	0.965	0.006
Ghrelin, pg/mL	0.989	0.981	0.998	0.015
Glucagon, pg/mL	1.024	0.987	1.061	0.202
Adiponectin, mcg/mL	0.991	0.977	1.006	0.237

Footnote: PAI-1-type 1 plasminogen activator inhibitor, IL-6-interleukin 6, MCP-1-monocytic chemoattractant protein type 1, TNFa-tumor necrosis factor-alpha, GIP-glucose-dependent insulinotropic polypeptide, PP-pancreatic polypeptide, OR-odds ratio.

**Table 7 biomolecules-12-01502-t007:** Results of logistic regression analysis of the association of adipokines with the risk of CB against the background of AO with standardization by sex, age and smoking status.

Parameter	Logistic Regression Analysis			
	OR	95% Confidence Interval (CI)		*p*
		Lower Bound	Upper Bound	
Smoking, smoking vs. not smoking	6.796	0.980	47.105	0.052
Adiponectin, mcg/mL	0.975	0.946	1.006	0.400
Adipsin, mcg/mL	0.957	0.864	1.060	0.991
Ghrelin, pg/mL	0.992	0.982	1.002	0.129
Glucagon, pg/mL	0.996	0.959	1.034	0.827
MCP-1, pg/mL	1.002	0.991	1.014	0.707
Lipocalin-2, mcg/mL	1.008	1.002	1.015	0.009
TNFa, pg/mL	1.715	0.985	2.988	0.057
GIP, pg/mL	1.046	1.001	1.096	0.041
Amylin, pg/mL	0.926	0.832	1.030	0.155

Footnote: MCP-1-monocytic chemoatractant protein-1, TNFa-tumor necrosis factor-alpha, GIP-glucose-dependent insulinotropic polypeptide, OR-odds ratio.

## Data Availability

Data of this study is a part of a larger study and it is available on special request.

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
