# Peer review of "Blood Adipokines/Cytokines in Young People with Chronic Bronchitis and Abdominal Obesity"

_biomolecules, 2022, doi:10.3390/biom12101502_

Round 1

Reviewer 1 Report

It is a manuscript where the objective was to evaluate the associations of some adipokines/cytokines and chronic bronchitis with the influence of abdominal obesity.

One strength of the study is the sample size.

In material and methods, the authors must describe why they take the cut-off point of physical activity as less than 3 hours per day.

In the material and methods section and in the results, they refer to CKD acronyms that are not found in any other part of the manuscript, which makes the reading and interpretation of these sections confusing.

In the results section, Table 1, they mention that the acronym for systolic blood pressure is SAD, but in the table they write it as SBP.

In lines 170-173, the authors report that they performed a logistic regression analysis and refer to table 3, but the figures do not match the table.

 The discussion is acceptable and they discuss what is in the literature on each of the adipokines/cytokines and chronic bronchitis.

In the conclusion section, the authors report that the results allow them to understand the pathogenesis of the development of inflammatory diseases of the bronchus in the setting of abdominal obesity. However, the reported associations of lipocalin-2, ghrelin, amylin and TNF alpha are not conclusive in logistic regression models, so it is very risky with these results to express that new methods of diagnosis and treatment could be developed in patients with chronic bronchitis.

Author Response

Point 1: In material and methods, the authors must describe why they take the cut-off point of physical activity as less than 3 hours per day.

 Response 1: A description of physical activity has been added to the Materials and Methods section. We have changed information in tables ("days" to "weeks").

Point 2: In the results section, Table 1, they mention that the acronym for systolic blood pressure is SAD, but in the table they write it as SBP.

Response 2: Appropriate corrections have been made

Point 3: In lines 170-173, the authors report that they performed a logistic regression analysis and refer to table 3, but the figures do not match the table.

Response 3: The title "Logistic regression analysis" has been added to table 3

Point 4: The discussion is acceptable and they discuss what is in the literature on each of the adipokines/cytokines and chronic bronchitis.

Response 4: Thank you very much for appreciating our article.

Point 5: In the conclusion section, the authors report that the results allow them to understand the pathogenesis of the development of inflammatory diseases of the bronchus in the setting of abdominal obesity. However, the reported associations of lipocalin-2, ghrelin, amylin and TNF alpha are not conclusive in logistic regression models, so it is very risky with these results to express that new methods of diagnosis and treatment could be developed in patients with chronic bronchitis.

 Response 5:  We certainly cannot state that lipocalin-2, ghrelin, amylin and TNF alpha should be used as diagnostic predictors. However, in the logistic regression model of associations of adipokines with CВ against the background of abdominal obesity, they performed well. We will continue our research in this area to confirm our assumptions.

Reviewer 2 Report

The manuscript by Dmitrievna et al examines the role of cytokines in young people with chronic bronchitis and abdominal obesity.

I have major concerns and my feedback is outlined below.

There is nor clear conclusion in the abstract, line- 24-25.

Certain abbreviations like PP should be abbreviated earlier in the text.

Discussion can be shortened since it loses flow of the manuscript and should specifically discuss the results.

How did the authors define age group 25-44 as young?

Also, conclusion in the end, 395-400 is very vague.

The results needs to explicitly discussed and what are the associations of these cytokines in relation to CD and AO. This is not clear at all.

Author Response

Point 1: There is nor clear conclusion in the abstract, line- 24-25.

 Response 1: For better understanding of the abstract, corrections have been made

Point 2: Certain abbreviations like PP should be abbreviated earlier in the text.

Response 2: All abbreviations are corrected.

Point 3: Discussion can be shortened since it loses flow of the manuscript and should specifically discuss the results.

 Response 3: Thank you very much for your comment. We believe that a significant discussion makes the perception of the article and its results more understandable to the reader. With the permission of the editors, we would not like to shorten this section.

Point 4: How did the authors define age group 25-44 as young?

 Response 4: Age group 25-44 is allocated according to the classification of the World Health Organization

Point 5: Also, conclusion in the end, 395-400 is very vague.

 Response 5: Given the design of this study, no more specific conclusions can be drawn. However, data on the association of some adipokines with chronic bronchitis associated with abdominal obesity will allow them to be included in cohort studies, from which we can draw a more specific conclusion.

Point 6: The results needs to explicitly discussed and what are the associations of these cytokines in relation to CD and AO. This is not clear at all.

 Response 6: Under the abbreviation CD, did you mean CB? If yes, then, аssociations of adipokines with chronic bronchitis, both with and without abdominal obesity, are presented in the results section in tables 6 and 7. In the discussion section, we tried to convey as fully as possible the putative mechanisms of changes in these adipocytes in chronic bronchitis.

Round 2

Reviewer 1 Report

It is a manuscript that has some errors since in the material and methods section and in the results they refer to CKD acronyms that are not in any other part of the manuscript, which makes the reading and interpretation of these sections of the manuscript confusing. The authors could define CKD for better interpretation..

Author Response

Point 1: It is a manuscript that has some errors since in the material and methods section and in the results they refer to CKD acronyms that are not in any other part of the manuscript, which makes the reading and interpretation of these sections of the manuscript confusing. The authors could define CKD for better interpretation.

 Response 1: We apologize for the typo in the text of the article. CKD corrected to CB.

Reviewer 2 Report

The authors have satisfactorily addressed some of the concerns.

Author Response

Пункт 1: Авторы удовлетворительно решили некоторые проблемы.

 Ответ 1: Спасибо, что оценили нашу работу
